☝️ PLOS | ONE

# Validation of risk factors for recurrence of renal cell carcinoma: Results from a large single-institution series

**Johannes C. van der Mijn**[1,2], **Bashir Al Hussein Al Awamlh**[3], **Aleem Islam Khan**[3], **Lina Posada-Calderon**[3], **Clara Oromendia**[4], **Jonathan Fainberg**[3], **Mark Alshak**[3], **Rahmi Elahjji**[3], **Hudson Pierce**[3], **Benjamin Taylor**[3], **Lorraine J. Gudas**[1], **David M. Nanus**[5], **Ana M. Molina**[5], **Joseph Del Pizzo**[3], **Douglas S. Scherr**[3]*

1 Department of Pharmacology, Weill Cornell Medicine, New York-Presbyterian Hospital, New York, NY, United States of America, 2 Department of Medical Oncology, Amsterdam University Medical Center, Amsterdam, The Netherlands, 3 Department of Urology, Weill Cornell Medicine, New York-Presbyterian Hospital, New York, NY, United States of America, 4 Department of Biostatistics and Epidemiology, Weill Cornell Medicine, New York-Presbyterian Hospital, New York, NY, United States of America, 5 Division of Hematology/Oncology, Department of Medicine, Weill Cornell Medicine, New York-Presbyterian Hospital, New York, NY, United States of America

* dss2001@med.cornell.edu

**Data Availability Statement:** Data cannot be shared publicly because of privacy regulations. Data are available from the WCM Institutional Data

## Abstract

### Purpose

To validate prognostic factors and determine the impact of obesity, hypertension, smoking and diabetes mellitus (DM) on risk of recurrence after surgery in patients with localized renal cell carcinoma (RCC).

### Materials and methods

We performed a retrospective cohort study among patients that underwent partial or radical nephrectomy at Weill Cornell Medicine for RCC and collected preoperative information on RCC risk factors, as well as pathological data. Cases were reviewed for radiographic evidence of RCC recurrence. A Cox proportional-hazards model was developed to determine the contribution of RCC risk factors to recurrence risk. Disease-free survival and overall survival were analyzed using the Kaplan-Meier method and log-rank test.

### Results

We identified 873 patients who underwent surgery for RCC between the years 2000–2015. In total 115 patients (13.2%) experienced a disease recurrence after a median follow up of 4.9 years. In multivariate analysis, increasing pathological T-stage (HR 1.429, 95% CI 1.265–1.614) and Nuclear grade (HR 2.376, 95% CI 1.734–3.255) were independently associated with RCC recurrence. In patients with T1-2 tumors, DM was identified as an additional independent risk factor for RCC recurrence (HR 2.744, 95% CI 1.343–5.605). Patients with DM had a significantly shorter median disease-free survival (1.5 years versus 2.6 years, p = 0.004), as well as median overall survival (4.1 years, versus 5.8 years, p<0.001).

Access / Ethics Committee (contact via mr. Thomas Flynn, thf3001@med.cornell.edu) for researchers who meet the criteria for access to confidential data and complete HIPPAA training. The study is registered under the title 'Identification of Prognostic Factors in Urological Malignancies' with protocol number 1403014960-01.

**Funding:** This study was supported by The Frederick J. and Theresa Dow Wallace Fund of the New York Community Trust, the Weiss family, the Turobiner Kidney Cancer Research Fund and partly by U54 CA210184 and Weill Cornell funds. The funder had no role in study design, data collection and analysis, decision to publish, or preparation of the manuscript.

**Competing interests:** The authors have declared that no competing interests exist.

## Conclusions

We validated high pathological T-stage and nuclear grade as independent risk factors for RCC recurrence following nephrectomy. DM is associated with an increased risk of recurrence among patients with early stage disease.

## Introduction

Renal cell carcinoma (RCC) is the most common neoplasm arising from the kidney cortex. Multiple histological subtypes exist with clear cell, papillary and chromophobe RCC accounting for 75%, 15% and 5% of RCCs, respectively[1,2]. Large cohort studies and meta-analyses have identified smoking, obesity and hypertension as the most important risk factors for the development of RCC[3,4]. In some studies type 2 diabetes mellitus (DM) has also been found to be independently associated with a risk of developing RCC. One meta-analysis revealed a 42% and 70% increased risk of developing RCC compared with non-diabetic men and women, respectively [5]. However, little is known about the role of these comorbidities after disease onset and during follow up.

Partial and radical nephrectomies are important treatments for patients with RCC, when the disease is confined to the kidney. In the majority of the patients, this treatment is curative with approximately 27% of the patients experiencing disease recurrence[6,7]. Two prognostic scoring systems are currently in use to estimate the recurrence risk of patients with RCC after surgery. The UCLA Integrated Scoring System (UISS) stratifies patients into three risk categories, based on pathological tumor stage (T-stage), Nuclear grade and ECOG performance status[8,9]. The SSIGN risk score incorporates tumor stage, size (>5cm), Nuclear grade and tumor necrosis into a risk score and was also found to be associated with recurrence of patients with clear cell RCC[10,11]. While these pathological features are rational risk factors for recurrence, few studies have investigated the impact of smoking, obesity, hypertension and DM on the recurrence risk following treatment of early stage RCC. We performed a comprehensive analysis of RCC risk factors and their association with recurrence after surgery for RCC.

## Materials and methods

### Patient cohort

A retrospective cohort study was conducted at our institution following approval by the Institutional Review Board at Weill Cornell Medicine (IRB approval #1403014960). All patients that were 18 years or older and underwent partial or radical nephrectomy with curative intent between January 2000 and January 2015 were included in the analysis. All data were analyzed anonymously. Patients that had a cytoreductive nephrectomy and were diagnosed with distant metastasis prior to surgery (n = 15) were not included in the analysis. Information about clinical parameters was registered as collected at the preoperative screening by the treating physician and included ASA score, gender, age, race, body weight, height, serum creatinine level, history of smoking and comorbidities, including hypertension, DM and/or dyslipidemia. The ASA classification score, as defined by the American Society of Anesthesiologist, was used as a measure for general health of the patient. Presence of comorbidities was registered according to the prior medical history provided by the referring physician or appropriate treating physician and when possible verified by use of comedication. Incidental laboratory or blood pressure measurements were not considered for a diagnosis of hypertension, DM and/or

dyslipidemia. Chronic renal insufficiency (CRI) was defined as a glomerular filtration rate (eGFR) <60 ml/min as estimated from serum creatinine levels according to the MDRD formula. All resection specimens were reviewed by a genitourinary pathologist according to routine clinical practice. The presence of malignancy, tumor stage, tumor histology, and other pathological features were determined according to the guidelines of the College of American Pathologists and recorded retrospectively. Thirteen patients were excluded because of missing data concerning tumor histology.

## Study outcomes

After surgery, patients were followed according to the guideline of the National Comprehensive Cancer Network (NCCN)[12]. This surveillance protocol comprises of history, physical examination, plasma creatinine, urinalysis and abdominal and chest CT/MRI imaging every 6–12 months until five years after surgery. Our primary outcome was RCC recurrence, either local or distant, deemed by treating surgeon or hematologist/oncologist, after a disease-free interval following surgery. The secondary outcome was to assess the role of metabolic factors in patients with early stage T1-2 tumors. The date of the first CT/MRI scan that showed evidence of disease recurrence was used as recurrence date. The disease-free survival (DFS) time was defined as the time from surgery to recurrence date.

## Statistical analysis

Statistical significance of differences in categorical values was assessed by using the $\chi^2$ (Chi-square) test and the Fisher's exact test. We examined the association between clinical variables and time to RCC recurrence with a Cox proportional hazard model. All factors that were statistically significant associated with recurrence in univariate analysis were included in the multivariate analysis. Hazard ratios (HRs) and 95% confidence intervals (CIs) were estimated from the models. To calculate DFS and overall survival the Kaplan-Meier survival analysis was used. We assessed the statistical significance of survival differences between groups by the log-rank test. A two-sided $p < 0.05$ was considered to indicate significance. All analyses were performed using SPSS version 25 (SPSS Inc., Chicago, IL).

## Results

A total of 873 patients were identified and included in this study. The clinical and demographic characteristics are presented in Table 1. The median age was 63 (interquartile range (IQR) of 53–71), 66.6% were male, and 66.7% Caucasian. Median BMI was 27.2 (IQR 24.4–30.7), and 29.1% of the patients were obese (defined as BMI >30). Overall, 14.3%, 24.3%, and 56.6% of the patients had DM, dyslipidemia and hypertension at the time of diagnosis, respectively. A total of 47.3% had a history of smoking and 12.1% had chronic renal insufficiency. A total of 1008 tumors were resected. A relatively large proportion of patients had a radical nephrectomy (45.9%), particularly in the years preceding 2004, when this was still the preferred surgical approach. The characteristics of these tumors are presented in Table 2. The large majority of the patients presented with clear cell RCC (56.6%). The remainder were diagnosed with papillary RCC (17.8%), chromophobe RCC (11.1%), oncocytoma (9.3%), and other histologies (5.3%). Seventy three percent were staged as T1 and 10.3% as T2. Very few patients had signs of lymph node metastasis (0.5%) at the time of surgery. The tumors comprised of 4.1%, 54.8%, 33.5% and 7.5% nuclear grade 1, 2, 3 and 4, respectively.

The median follow up of the cohort was 4.9 years (IQR 1.3–9.7 years). In total, 115 patients (13.2%) experienced a disease recurrence with a median time to relapse of 2.4 years (IQR 0.8–5.3 years). The disease characteristics of patients with a recurrence are specified in Table 2.

**Table 1. RCC risk factor profile of 873 patients undergoing surgical resection of a renal tumor.**

| Characteristic | Patients |
|---|---|
| **Age** | |
| Median (IQR) | 63 (53–71) |
| **Gender**, N (%) | |
| M | 581 (66.6) |
| F | 292 (33.4) |
| **Race**, N (%) | |
| Caucasian | 582 (66.7) |
| African American | 66 (7.6) |
| Asian | 31 (3.6) |
| Hispanic | 31 (3.6) |
| Unknown | 143 (16.4) |
| **Surgery, N (%)** | |
| Partial nephrectomy | 469 (53.7) |
| Radical nephrectomy | 402 (46.0) |
| **ASA score**, N (%) | |
| I | 33 (3.8) |
| II | 508 (58.2) |
| III | 295 (33.8) |
| IV | 25 (2.9) |
| **BMI** | |
| Median (IQR) | 27.2 (24.4–30.7) |
| **BMI category**, N (%) | |
| <25 | 245 (30.2) |
| 25–30 | 329 (40.6) |
| >30 | 236 (29.1) |
| **Smoking**, N (%) | 408 (47.3) |
| **Diabetes mellitus**, N (%) | 124 (14.3) |
| **Dyslipidemia**, N (%) | 212 (24.3) |
| **Hypertension**, N (%) | 492 (56.6) |
| **Renal insufficiency,** N (%) | 104 (12.1) |

Patients with a disease recurrence were more frequently diagnosed with higher T-stage and positive nodal clear cell RCC for which they more often received radical nephrectomy. We detected no significant difference in the frequency of positive surgical margins after the initial surgery in patients with a recurrence. In the unadjusted prediction model for recurrence, female gender (HR 0.504, 95% CI 0.321–0.790), partial nephrectomy (HR 0.375, 95% CI 0.255–0.551), ASA score (HR 1.556, 95% CI 1.162–2.083), pathological T-stage (HR 1.742, 95% CI 1.585–1.915), Nuclear grade (HR 3.943, 95% CI 2.946–5.277), clear cell histology (HR 2.370, 95% CI 1.563–3.595), DM (HR 1.951, 95% CI 1.233–3.088), and hypertension (HR 1.815, 95% CI 1.223–2.692) were associated with RCC recurrence (Table 3). In the multivariate analysis, including all patients, only T-stage (HR 1.429, 95% CI 1.265–1.614) and Nuclear grade (HR 2.376, 95% CI 1.734–3.255) were significantly associated with RCC recurrence. To investigate the role of metabolic factors in patients with early stage tumors, we performed a multivariate analysis of only patients with T1-2 tumors. Female gender (HR 0.409, 95% CI 0.198–0.848), DM (HR 2.744, 95% CI 1.343–5.605), T-stage (HR 1.601, 95% CI 1.186–2.161) and Nuclear grade (HR 2.429, 95% CI 1.524–3.872) were significantly associated with recurrence. We next investigated the impact of DM on the length of the disease-free survival (DFS)

**Table 2. Renal tumor characteristics of all patients and patients that developed a recurrence after follow up.**

| Characteristic | All patients | Recurrence | P |
|---|---|---|---|
| **Histology, N (%)** | | | <0.001 |
| Clear cell | 494 (56.6) | 85 (73.9) | |
| Papillary | 155 (17.8) | 12 (10.4) | |
| Chromophobe | 97 (11.1) | 6 (5.2) | |
| Oncocytoma | 81 (9.3) | 3 (2.6) | |
| Other | 46 (5.3) | 9 (7.8) | |
| **T-stage, N (%)** | | | <0.001 |
| T1 | 633 (73.6) | 44 (39.2) | |
| T2 | 89 (10.3) | 17 (15.1) | |
| T3 | 134 (15.6) | 48 (42.9) | |
| T4 | 5 (0.6) | 3 (2.6) | |
| **Nodal stage, N (%)** | | | <0.001 |
| Nx | 788 (91.4) | 91 (81.3) | |
| pN0 | 69 (8.0) | 17 (15.2) | |
| pN1 | 5 (0.5) | 4 (3.6) | |
| **Nuclear grade, N (%)** | | | <0.001 |
| 1 | 26 (4.1) | 1 (1.0) | |
| 2 | 345 (54.8) | 31 (30.7) | |
| 3 | 211 (33.5) | 43 (42.6) | |
| 4 | 47 (7.5) | 26 (25.7) | |
| **Surgery, N (%)** | | | <0.001 |
| Partial | 468 (53.9) | 37 (32.2) | |
| Radical | 399 (45.9) | 76 (67.0) | |
| **Surgical margins, N (%)** | | | 0.264 |
| Negative | 790 (92.0) | 100 (89) | |
| Positive | 69 (8.0) | 12 (11) | |
| **Location of recurrence, N (%)** | | | - |
| Local | - | 16 (14) | |
| Distant | - | 99 (86) | |

and overall survival (Fig 1A). The median time to relapse for patients with DM was 1.5 years, versus a median of 2.6 years for patients without DM (p = 0.004). For patients with T1-2 tumors and DM, the median time to recurrence was 1.6 years, versus a median of 4.8 years among patients without DM (p = 0.022). Similarly, we noted significant differences in overall survival with patients with DM having a median overall survival of 4.1 years, versus a median of 5.8 years for patients without DM (Fig 1B). Collectively, these results indicate that DM is associated with an increased risk of recurrence and a shorter disease-free interval, particularly among patients with early stage RCC.

## Discussion

Previous risk stratification models identified pathological T-stage and Nuclear grade as important risk factors for recurrence. We here confirmed that these factors are the most powerful clinical predictors of recurrence, with increasing tumor size and higher nuclear grade associated with an increasing risk in a large cohort of patients. Importantly, this association with recurrence is independent of tumor histology, type of surgery and metabolic risk factors. Our analysis focused particularly on metabolic risk factors, since these have been identified as

**Table 3. Cox regression analysis of risk factors for RCC recurrence among all patients and patients with T1-2 tumors.**

| | Unadjusted Models | | | Multivariate analysis | | | | | |
| --- | --- | --- | --- | --- | --- | --- | --- | --- | --- |
| | | | | All patients | | | Patients with T1-2 tumors | | |
| Variable | HR | 95% CI | *P* | HR | 95% CI | *P* | HR | 95% CI | *P* |
| Age | 1.003 | 0.988–1.018 | 0.732 | | | | | | |
| Gender (F vs. M) | 0.504 | 0.321–0.790 | **0.003** | 0.651 | 0.395–1.074 | 0.093 | 0.409 | 0.198–0.848 | **0.016** |
| Race (Caucasian) | 1.023 | 0.968–1.081 | 0.425 | | | | | | |
| Partial nephrectomy | 0.375 | 0.255–0.551 | <**0.001** | 0.677 | 0.424–1.083 | 0.104 | 1.023 | 0.553–1.890 | 0.943 |
| ASA score | 1.556 | 1.162–2.083 | **0.003** | 0.851 | 0.613–1.180 | 0.333 | 0.737 | 0.465–1.169 | 0.195 |
| T stage | 1.742 | 1.585–1.915 | <**0.001** | 1.429 | 1.265–1.614 | <**0.001** | 1.601 | 1.186–2.161 | **0.002** |
| T1-2 | ref | - | - | | | | | | |
| T3-4 | 7.752 | 5.301–11.335 | <**0.001** | | | | | | |
| Nuclear grade | 3.943 | 2.946–5.277 | <**0.001** | 2.376 | 1.734–3.255 | <**0.001** | 2.429 | 1.524–3.872 | <**0.001** |
| Clear cell histology | 2.370 | 1.563–3.595 | <**0.001** | 1.199 | 0.703–2.044 | 0.504 | 1.467 | 0.707–3.044 | 0.304 |
| BMI (continuous) | 0.970 | 0.936–1.005 | 0.090 | | | | | | |
| Obesity (BMI>30) | 0.721 | 0.464–1.119 | 0.145 | | | | | | |
| Smoking | 1.159 | 0.804–1.670 | 0.430 | | | | | | |
| Diabetes Mellitus | 1.951 | 1.233–3.088 | **0.004** | 1.614 | 0.978–2.662 | 0.061 | 2.744 | 1.343–5.605 | **0.006** |
| Dyslipidemia | 1.321 | 0.794–2.198 | 0.283 | | | | | | |
| Hypertension | 1.815 | 1.223–2.692 | **0.003** | 1.512 | 0.957–2.390 | 0.077 | 1.836 | 0.988–3.414 | 0.055 |
| Renal insufficiency | 0.991 | 0.544–1.804 | 0.976 | | | | | | |

dominant risk factors for development of kidney cancer in general. Despite the strong association with disease onset, we noted that these factors have a minor role during follow up, particularly in patients with (locally) advanced disease. In line with numerous previous studies [13,14], we did detect a notable association between a medical history of DM and an increased risk of RCC recurrence. This association was only statistically significant in patients with early stage disease (T1-2 tumors). We also detected an association between DM and tumor stage, with diabetic patients frequently having higher stage tumors, and a shorter disease-free survival interval. Collectively, these results suggest that DM may promote RCC progression or that tumors from patients with DM are more aggressive. No information was available about co-medication use and glycemic control of our cohort, which is a limitation of our study and limits the scope of these findings.

In this study, we detected no associations between RCC recurrence and hypertension, obesity, dyslipidemia, chronic renal insufficiency, and smoking in our multivariate analyses. Particularly, abdominal obesity is associated with insulin resistance, vascular endothelial dysfunction and an abnormal lipid profile, ultimately leading to hyperglycemia and hyperinsulinemia. Increased insulin-like growth factor receptor 1 (IGF1R) has been associated with poor disease specific survival of patients with early stage RCC[15]. We hence speculate that hyperinsulinemia, that is associated with type 2 diabetes mellitus, promotes RCC progression through enhanced signaling of IGF1R and PI3K in cancer cells[16]. Alternatively, persistent hyperglycemia and poor glycemic control may directly fuel RCC tumors, which are known to heavily rely on aerobic glycolysis for proliferation[17]. We believe that more research is warranted to elucidate the role of these individual factors during disease progression.

Our results are consistent with previously developed nomograms, such as the UISS, which included T-stage, Nuclear grade, and ECOG performance status[8,9]. No information was available on the ECOG performance status of the patients in our cohort, but ASA score was not significantly associated with RCC recurrence in our multivariate analysis. Our patient

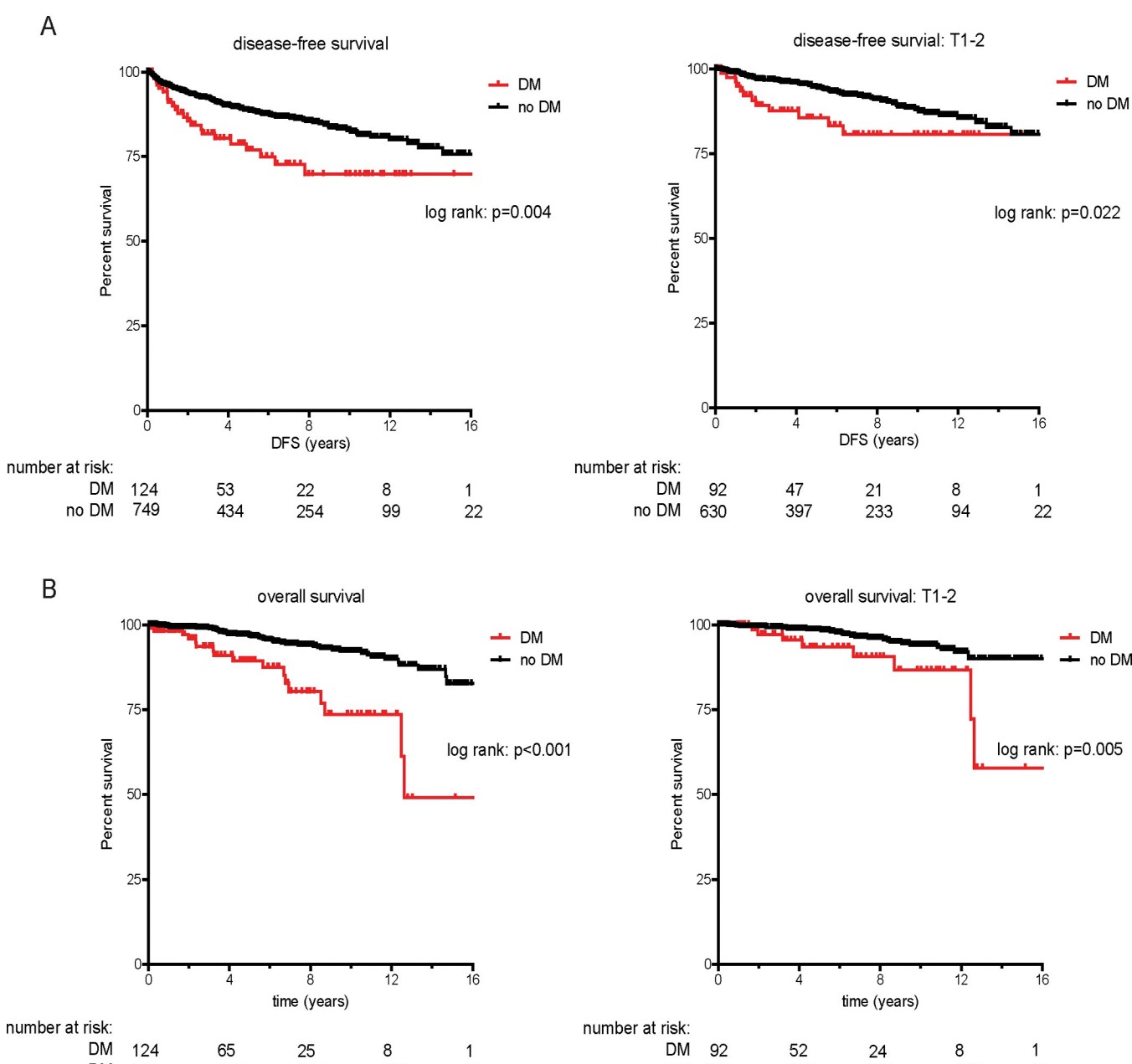

**Fig 1.** Impact of diabetes mellitus on disease-free survival (DFS, A) and overall survival (OS, B) after surgery among all patients and patients with T1-2 tumors.

cohort contained a higher frequency of T1-2 stage tumors compared to previous cohorts, which likely contributed to an overall better performance status, ASA score and an overall lower recurrence rate as compared to previous studies (13.4% versus 27.6%)[6,7]. Future research will have to show whether it is possible to simplify the UISS nomogram by removing the ECOG performance status. Further refinement of prognostic nomograms for patients with RCC may come from molecular profiling studies. Previous research showed that the 'Cell Cycle Proliferation (CCP)' and 'ClearCode34' RNA expression profiles in RCC tumors may improve the prognostic classification by UISS of patients with localized RCC[18,19]. Interestingly, few studies have looked at genomic markers. Extensive genomic profiling of primary

RCC tumors has been performed by the TCGA, showing correlations, for example, between BAP1 mutations and prognosis in patients with clear cell RCC[20]. Smaller tumors, such as predominantly seen in our cohort, have a reduced genomic complexity with fewer subclonal events[21]. Some of the subclonal genomic events described in early stage tumors, such as somatic copy number loss of chromosome 9p and 14q, were recently found to be associated with the development of metastatic disease[22]. These findings suggest that in some patients the metastatic potential of tumors is determined at an early stage. More research is needed to determine the role of such genomic markers in addition to the clinical factors such as DM. In conclusion, here we validated that pathological T-stage, and Nuclear grade are independent risk factors for RCC recurrence. In patients with early stage RCC, DM was an additional independent risk factor for RCC recurrence. Prospective research is needed to further elucidate the role of DM in the development and progression of RCC.

## Conclusion

Pathological T-stage and nuclear grading are the most powerful clinical predictors of RCC recurrence following nephrectomy. DM is associated with an increased risk of recurrence among patients with early stage disease.

## Acknowledgments

We thank Thomas Flynn for administrative assistance and the members of the Gudas lab for stimulating discussions and valuable input.

## Author Contributions

**Conceptualization:** Johannes C. van der Mijn, Bashir Al Hussein Al Awamlh, Lorraine J. Gudas, David M. Nanus, Ana M. Molina, Douglas S. Scherr.

**Data curation:** Bashir Al Hussein Al Awamlh, Aleem Islam Khan, Lina Posada-Calderon, Clara Oromendia, Jonathan Fainberg.

**Formal analysis:** Bashir Al Hussein Al Awamlh, Aleem Islam Khan, Lina Posada-Calderon, Douglas S. Scherr.

**Investigation:** Aleem Islam Khan, Lina Posada-Calderon, Jonathan Fainberg, Mark Alshak, Rahmi Elahjji, Hudson Pierce, Benjamin Taylor, David M. Nanus, Joseph Del Pizzo, Douglas S. Scherr.

**Methodology:** Johannes C. van der Mijn, Bashir Al Hussein Al Awamlh, Aleem Islam Khan, Lina Posada-Calderon, Clara Oromendia, Jonathan Fainberg, David M. Nanus, Ana M. Molina, Douglas S. Scherr.

**Project administration:** Joseph Del Pizzo, Douglas S. Scherr.

**Resources:** Lorraine J. Gudas, Ana M. Molina, Joseph Del Pizzo, Douglas S. Scherr.

**Software:** Lorraine J. Gudas.

**Supervision:** Johannes C. van der Mijn, Clara Oromendia, Lorraine J. Gudas, David M. Nanus, Ana M. Molina, Joseph Del Pizzo, Douglas S. Scherr.

**Writing – original draft:** Johannes C. van der Mijn, Bashir Al Hussein Al Awamlh, David M. Nanus, Douglas S. Scherr.

**Writing – review & editing:** Johannes C. van der Mijn, Bashir Al Hussein Al Awamlh, Aleem Islam Khan, Lina Posada-Calderon, Clara Oromendia, Jonathan Fainberg, Mark Alshak,

Rahmi Elahjji, Hudson Pierce, Benjamin Taylor, Lorraine J. Gudas, David M. Nanus, Ana M. Molina, Joseph Del Pizzo, Douglas S. Scherr.

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
