## [Decision Letter · Decision Letter 0]

29 Oct 2019

PONE-D-19-21440

Validation of risk factors for recurrence of renal cell carcinoma: results from a large single-institution series

PLOS ONE

Dear Dr. van der Mijn,

Thank you for submitting your manuscript to PLOS ONE. After careful consideration, we feel that it has merit but does not fully meet PLOS ONE’s publication criteria as it currently stands. Therefore, we invite you to submit a revised version of the manuscript that addresses the points raised during the review process, specifically the comments raised by Reviewer #2.

We would appreciate receiving your revised manuscript by Dec 13 2019 11:59PM. To enhance the reproducibility of your results, we recommend that if applicable you deposit your laboratory protocols in protocols.io, where a protocol can be assigned its own identifier (DOI) such that it can be cited independently in the future. For instructions see: http://journals.plos.org/plosone/s/submission-guidelines#loc-laboratory-protocols

We look forward to receiving your revised manuscript.

Kind regards,

Donald P. Bottaro

Academic Editor

PLOS ONE

Journal Requirements:

Reviewers' comments:

Reviewer's Responses to Questions

**Comments to the Author**

1. Is the manuscript technically sound, and do the data support the conclusions?

Reviewer #1: Yes

Reviewer #2: Yes

2. Has the statistical analysis been performed appropriately and rigorously? 

Reviewer #1: Yes

Reviewer #2: Yes

3. Have the authors made all data underlying the findings in their manuscript fully available?

Reviewer #1: Yes

Reviewer #2: Yes

4. Is the manuscript presented in an intelligible fashion and written in standard English?

Reviewer #1: Yes

Reviewer #2: Yes

5. Review Comments to the Author

Reviewer #1: The authors present an analysis risk of recurrence of a contemporary single institution series of surgically managed localized RCC. They have looked at traditional demographic and pathologic risk factors including age, gender, race, smoking statusstage, histology and grade. The author have additionally focused on the contribution of components of metabolic syndrome (DM, HTN) and other medical conditions to the risk of recurrence. While the contributions of these factors to the development of RCC have been explored previously, they have not been incorporated into other more widely used risk stratification systems (SSIGN, UISS). The authors have analyzed these factors and found that for patients with T1-T2 tumors diabetes was an even stronger predictor of recurrence that stage and grade. The author provide some potential mechanisms for this (increased IGF signaling and increased hyperglycemia fueling aerobic glycolysis). This finding will need to be validated in other studies as it is surprising that DM would have a seemingly more powerful effect than grade, even when limited to T1-T2 tumors.

This a well-written manuscript which looks at less explored factors for recurrence (DM, HTN and other metabolic factors). The data is well-interpreted and supports the conclusions. The introduction and discussion frame the data and provide good context.

Reviewer #2: This is a very interesting manuscript

Please provide if possible answers to the following queries

1- Was a certain group of patients with early stage disease surveilled before proceeding to surgery? If so, what was the rate of recurrence in these patients?

2- How do you explain the high rate of radical nephrectomies in your series (almost 50%) despite the fact that 73% were T1 (should you discuss possibly a high prevalence of T1b patients and how it may influence your outcome analysis?)

3- In your statistical methods, you mention how you analyse categorical variables but no clear mention is made about continuous variable; In table 3, continuous variables are included: how were these analyzed? Why was stage considered continuous?

4- In table 2, I am confused at the line with positive margins. It seems that 69 patients had positive margins ''of which'' 100 patients recurred? Please clarify

5- Finally, can you discuss whether you plan confirming your finding with an external cohort and what the next steps are before DM may be added as a risk factor to present nomograms as a practice changer.

6. PLOS authors have the option to publish the peer review history of their article (what does this mean?). If published, this will include your full peer review and any attached files.

Reviewer #1: No

Reviewer #2: No

---

## [Author Response · Author response to Decision Letter 0]

15 Nov 2019

Reviewer #1: The authors present an analysis risk of recurrence of a contemporary single institution series of surgically managed localized RCC. They have looked at traditional demographic and pathologic risk factors including age, gender, race, smoking statusstage, histology and grade. The author have additionally focused on the contribution of components of metabolic syndrome (DM, HTN) and other medical conditions to the risk of recurrence. While the contributions of these factors to the development of RCC have been explored previously, they have not been incorporated into other more widely used risk stratification systems (SSIGN, UISS). The authors have analyzed these factors and found that for patients with T1-T2 tumors diabetes was an even stronger predictor of recurrence that stage and grade. The author provide some potential mechanisms for this (increased IGF signaling and increased hyperglycemia fueling aerobic glycolysis). This finding will need to be validated in other studies as it is surprising that DM would have a seemingly more powerful effect than grade, even when limited to T1-T2 tumors.

This a well-written manuscript which looks at less explored factors for recurrence (DM, HTN and other metabolic factors). The data is well-interpreted and supports the conclusions. The introduction and discussion frame the data and provide good context.

AUTHOR REPLY: We thank the reviewer for critically reading our manuscript and his feedback. 

Reviewer #2: This is a very interesting manuscript

Please provide if possible answers to the following queries

1- Was a certain group of patients with early stage disease surveilled before proceeding to surgery? If so, what was the rate of recurrence in these patients?

AUTHOR REPLY: We are grateful for the feedback the reviewer has provided. Our database only included patients who underwent surgery. Unfortunately, data on the disease course prior to surgery was not collected. 

2- How do you explain the high rate of radical nephrectomies in your series (almost 50%) despite the fact that 73% were T1 (should you discuss possibly a high prevalence of T1b patients and how it may influence your outcome analysis?)

AUTHOR REPLY: The first patients in our cohort received surgery in early 2000. Although nephron-sparing surgery was recognized as potential beneficial treatment strategy in selected patients at that time, mature survival data showing non-inferiority of this treatment approach, particularly in T1b tumors, were not published earlier than 2004. In line with evolving practice, we observed a decrease in the proportion (116/222 pts, 52%) of patients with T1 tumors that underwent radical nephrectomy after 2004 compared to before (101/411 pts, 25%). Concurrently, we noted an increase in the partial nephrectomy rates in the years following 2004 (48% to 75%), as it was widely recognized as a safe treatment and hence was rapidly adopted as preferred treatment modality at our institute. We added this observation to the results section of the manuscript on page 8. 

3- In your statistical methods, you mention how you analyse categorical variables but no clear mention is made about continuous variable; In table 3, continuous variables are included: how were these analyzed? Why was stage considered continuous?

AUTHOR REPLY: The data analyzed in our study were all categorical, no results from continuous data were included in the tables. Pathological T-stage is also an example of an (ordinal) categorical variable. This has been corrected in table 3. 

4- In table 2, I am confused at the line with positive margins. It seems that 69 patients had positive margins ''of which'' 100 patients recurred? Please clarify

AUTHOR REPLY: We thank the reviewer for critically reviewing our results. In total, 69 patients had positive surgical margins, of which 12 (17 %) experienced disease recurrence. Among patients with negative surgical margins (n=790), 100 (13%) had disease relapse. These numbers have now been modified in table 2.

5- Finally, can you discuss whether you plan confirming your finding with an external cohort and what the next steps are before DM may be added as a risk factor to present nomograms as a practice changer.

AUTHOR REPLY: In order to include DM in current prognostic nomograms for RCC recurrence additional prospective results are needed. We believe that our findings regarding the role of diabetes mellitus in RCC recurrence, warrant evaluation in a newly designed clinical study. To this end, we are planning a prospective multicenter observational clinical study in which patients with and without type 2 DM are followed after surgery. This study will include evaluation of co-medications, glycemic control and systemic endogenous insulin levels (c-peptide measurements), in addition to pathological T-stage and nuclear grade, to determine the mechanism by which DM might influence disease progression.

---

## [Editor Report · Decision Letter 1]

25 Nov 2019

Validation of risk factors for recurrence of renal cell carcinoma: results from a large single-institution series

PONE-D-19-21440R1

Dear Dr. van der Mijn,

We are pleased to inform you that your manuscript has been judged scientifically suitable for publication and will be formally accepted for publication once it complies with all outstanding technical requirements.

With kind regards,

Donald P. Bottaro

Academic Editor

PLOS ONE
---

## [Editor Report · Acceptance letter]

2 Dec 2019

PONE-D-19-21440R1 

Validation of risk factors for recurrence of renal cell carcinoma: results from a large single-institution series 

Dear Dr. van der Mijn:

I am pleased to inform you that your manuscript has been deemed suitable for publication in PLOS ONE. Congratulations! Your manuscript is now with our production department. 

With kind regards,

on behalf of

Dr. Donald P. Bottaro 

Academic Editor

PLOS ONE